# Potential Anti-Tuberculosis Activity of the Extracts and Their Active Components of *Anogeissus leiocarpa* (DC.) Guill. and Perr. with Special Emphasis on Polyphenols

**DOI:** 10.3390/antibiotics9070364

**Published:** 2020-06-29

**Authors:** Enass Y. A. Salih, Riitta Julkunen-Tiitto, Olavi Luukkanen, Marketta Sipi, Mustafa K. M. Fahmi, Pia Johanna Fyhrquist

**Affiliations:** 1Faculty of Pharmacy, Division of Pharmaceutical Biosciences, PO Box 56, University of Helsinki, FI-00014 Helsinki, Finland; pia.fyhrquist@helsinki.fi; 2Department of Forest Products and Industries, Faculty of Forestry, PO Box 13314, University of Khartoum, Khartoum 11111, Sudan; 3Viikki Tropical Resources Institute (VITRI), Department of Forest Sciences, PO Box 27, University of Helsinki, FI-00014 Helsinki, Finland; olavi.luukkanen@helsinki.fi (O.L.); or mkfahmi@uofk.edu (M.K.M.F.); 4Department of Environmental and Biological sciences, Faculty of Science and Forestry, University of Eastern Finland, 80101 Joensuu, Finland; riitta.julkunen-tiitto@uef.fi; 5Department of Forest Sciences, Faculty of Agriculture and Forestry, PO Box 27, University of Helsinki, FI-00014 Helsinki, Finland; marketta.sipi@helsinki.fi; 6Department of Forest Managements, Faculty of Forestry, University of Khartoum, PO Box 13314, Khartoum 11111, Sudan

**Keywords:** *Anogeissus leiocarpa*, Africa, tuberculosis, ellagitannins, ellagic acid derivatives, flavonoids, stilbenes, *Mycobacterium smegmatis*

## Abstract

In Sudanese traditional medicine, decoctions of the stem bark of *Anogeissus leiocarpa* are used for the treatment of tuberculosis (TB). However, this plant has not been investigated before for its antimycobacterial effects. Our screening results show, for the first time, that many extracts of various parts of *A. leiocarpa* exhibit growth inhibitory activity against *Mycobacterium smegmatis*. Minimum inhibitory concentration (MIC) values ranged between 625 and 5000 µg/mL, with an ethyl acetate extract of the root showing the lowest MIC value. The good antimycobacterial effects of the root part could be due to its high concentration of ellagic acid derivatives, ellagitannins, and flavonoids. Thin layer chromatography (TLC) fractionation resulted in some fractions with better activity than the starting point crude methanol extract (MIC 2500 µg/mL). Those fractions with the lowest MIC values contained a high number of antioxidant compounds. Fractions **3** and **4** (MIC 1500 and 1000 µg/mL, respectively) contained high concentrations of di-methyl ellagic acid ([M-H]^−^ 329.0318). Fraction **6** (MIC 2000 µg/mL) contained a lower concentration of di-methyl ellagic acid and was not as growth inhibitory as fractions **3** and **4**. Moreover, in fraction **3**, an acetylated ellagic acid derivative ([M-H]^−^ 343.0477) and di-methyl-ellagic acid xyloside ([M-H]^−^ 461.0739) were tentatively characterized. Di-methyl ellagic acid xyloside was also present in fraction **4** and could strongly contribute to the antimycobacterial effect of this fraction. Additionally, protocatechuic acid ([M-H]^−^ at m/z 153.0196) was present in fraction **4.** Our antimycobacterial results obtained from this research justify the use of *A. leiocarpa* in Sudanese folk medicine against cough related to TB. Roots, stem bark, and leaves of *A. leiocarpa* are sources for new potent anti-TB drug lead compounds.

## 1. Introduction

Tuberculosis (TB) is an infectious disease that mostly affects the lungs, but it also affects other organs such as bones, kidneys, or the brain [1]. *Mycobacterium tuberculosis*, the main causative bacterium of TB, is spread through cough, talk, and/or sneeze. However, age, sex, and socio-economic environment (poverty), as well as immunosuppression (HIV-infection), are considered as major factors affecting TB prevalence [1,2]. Approximately one third of the world population is estimated to be infected by TB, with most infections in a latent phase [3,4], and most of the new global TB cases are localized in Africa [5]. In 2019, the World Health Organization (WHO) [6] declared that 484,000 of the new global TB cases were resistant to the first-line antibiotic rifampicin, and 78% of them developed multidrug-resistant tuberculosis (MDR-TB). The new TB cases in 2018 were predominantly found among males, with a total of 5.4 million new cases, followed by 3.2 million new TB cases among females and 1.0 million new TB cases among children [5]. Moreover, at present, 1.0 million children die annually from MDR-TB [3,4]. According to the WHO, the global TB situation needs a focused effort and emergency action, with an estimated total cost of US $3.3 billion dollars/year to slow down the spread of the TB and to hinder the incidence of new resistant strains [1].

In some African countries with high poverty levels, a high number of the deaths among the poor people result from TB [7]. This is due to the scarcity of new and advanced medical inspection equipment, as well as laboratories lacking diagnostic equipment, high tariffs for TB drugs (US $25/person), and low income levels (less than US $1 per person/day) [8]. Thus, in many African countries, a high number of the TB patients use traditional medicine, including plant-based medicines for the treatment of prolonged cough and other symptoms related to TB. 

In Africa, many plant species belonging to the family Combretaceae are used customarily against cough and respiratory infections, including prolonged cough [9,10,11,12]. *Anogeissus leiocarpa*, the African birch (in Sudanese: Al-Sahab الصهب) is an example of these plants. *A. leiocarpa* occurs naturally in Central, East, and West African countries (Figure 1). According to African plant database and plant list database 2010 [13], there are nine species belonging to the genus *Anogeissus*, of which eight are native to tropical Asia. *A. leiocarpa* is the only African species, and it grows in various geographical zones and different vegetation types such as bushlands, savannas, and woodlands, and due to its good tolerance to drought and high soil salinity, it can even grow in drylands and arid and semi-arid grasslands [9,14,15]. *A. leiocarpa* is a deciduous tree that can reach heights of 15–18 m (Figure 1B). The bark is greyish and scaly, and the branches are drooping. The inflorescences are globose heads with yellow flowers, and the fruits are winged [16] (Figure 1C). In Sudanese traditional medicine, the stem bark is made into a decoction for the treatment of cough [17]. Additionally, there are ethno-pharmacological reports on the specific use of *A. leiocarpa* against tuberculosis, including symptoms such as bloody and prolonged cough [18,19]. Other uses in African traditional medicine of decoctions and macerations of *A. leiocarpa* include a broad spectrum of infectious diseases such as bacterial and fungal infections, as well as symptoms related to these infections, such as diarrhea and skin rashes [18,20,21].

Comparatively little research has been performed to verify the ethno-pharmacologically claimed antimicrobial effects of *A. leiocarpa*, and even fewer publications have dealt with the phytochemistry of this species (Figure 2). However, in accordance with the traditional medicinal uses of *A. leiocarpa*, some researchers have demonstrated significant in vitro antibacterial effects of its extracts against some bacterial strains, such as *Staphylococcus aureus*, *Pseudomonas aeruginosa*, *Shigella dysenteriae*, *Escherichia coli,* and *Klebsiella pneumoniae* (Figure 2) [18,22,23,24,25]. 

Phytochemical studies of extracts from *A. leiocarpa* have mainly reported on polyphenolic and phenolic compounds (Figure 3). Among these compound groups, we and other researchers have reported on the occurrence of a number of ellagitannins, ellagic acid and its derivatives, flavellagic acid, gallic acid and its derivatives, gallotannins, epicatechin-3-*O*-gallate, and epicatechin [18,26,27,28,29,30] (Figure 3). In addition, a number of flavonoids have been reported to occur in *A. leiocarpa* (Figure 3) [16,18,26,31,32,33,34].

Till today, only one earlier study investigated the antimycobacterial effects of *A. leiocarpa* using the Bacillus Calmette-Guérin tuberculosis vaccine containing a live attenuated strain of *Mycobacterium bovis* [20]. However, to the best of our knowledge, *A. leiocarpa* extracts have not been tested before against *Mycobacterium smegmatis*. *M. smegmatis* is known to have a high genetic homology with *M. tuberculosis*, as well as being rather resistant against rifampicin, and it is thus considered to be a representative organism to use in antimycobacterial testing [35,36]. Therefore, in this current paper, we present our research that aimed to validate the African traditional medicinal use of water extracts (macerations) and decoctions of *A. leiocarpa* against tuberculosis and its symptoms, such as cough, using *M. smegmatis* as the test bacterium. This paper presents the antimycobacterial results of a number of extracts and fractions of various polarities from the stem bark, roots, and leaves of *A. leiocarpa*. Additionally, high performance liquid chromatography with a diode array detector (HPLC-DAD) and ultra-high performance liquid chromatography quadrupole time of flight mass spectrometry (UHPLC/QTOF-MS) were used for the phytochemical profiling of an antimycobacterial root extract and its preparative reversed phase 18 (RP_18_)-thin layer chromatography (TLC) fractions. Preparative RP_18_-TLC was used to trace the portion of the root extract that could be responsible for its good antimycobacterial effects. 

## 2. Results

### 2.1. Extracts and Their Antimycobacterial Effects 

Thirty-nine extracts, four thin layer chromatography fractions, and five pure compounds known to occur in *A. leiocarpa* were tested for their growth inhibitory effects on *M. smegmatis*. The results are presented in Table 1 and Table 2. 

An ethyl acetate extract of the roots of *A. leiocarpa* showed the lowest MIC value (625 µg/mL) of all extracts in this study and a large inhibition zone diameter (IZD) of 28.50 mm; additionally, the growth inhibitory effects of this extract were dose-dependent.

Since a Soxhlet methanol extract of the roots also gave good growth inhibitory results (IZD 21.33 mm), an attempt was made to find the antimycobacterial components or compound combinations in this extract using RP_18_-TLC fractionation (Figure 4). Accordingly, the thin-layer fractions **Fr 3** (R_f_ 0.09), **Fr 4** (R_f_ 0.15), and **Fr 6** (R_f_ 0.27), all enriched with ellagic acid derivatives, showed better growth inhibition than the crude extract (MIC 2500 µg/mL), with MIC values of 1500, 1000, and 2000 µg/mL, respectively (Table 2). In comparison, we found that pure ellagic acid gave an MIC value of 500 µg/mL (Table 2). The most polar TLC fraction, **Fr 8** (R_f_ 0.45) showed a weaker antimycobacterial effect than the crude extract, with an MIC value of 3000 µg/mL (Table 2). 

As shown in Table 1 and Table 2, the hydrophilic extracts of the leaves gave good antimycobacterial effects in general, such that an ethyl acetate extract gave the largest IZD of 30.50 mm and an MIC of 2500 µg/mL, followed by a cold water extract (IZD 26.67 mm and MIC 2500 µg/mL), an aqueous fraction (IZD 24.67 mm), a methanolic Soxhlet extract (IZD 24.17 mm), and a hot water decoction (IZD 20.33 mm). 

Moreover, ethyl acetate extracts of the stem bark and stem wood of *A. leiocarpa* showed large diameters of inhibition zones against the growth of *M. smegmatis* (IZD 28.67 and 25.17 mm, respectively, and an MIC value of 2500 µg/mL for the ethyl acetate fraction of the stem bark). 

In contrast to the whole root, only weak or no growth inhibition was observed for the root bark extracts of *A. leiocarpa* (Table 1). 

In contrast to the more polar extracts, less polar or non-polar extracts such as dichloromethane and n-hexane extracts of the various parts of *A. leiocarpa* mostly showed weak or no growth inhibition of *M. smegmatis* (Table 1). This result was in accordance with previous research that non-polar extracts of *A. leiocarpa* were not active against a number of strains of gram-positive bacteria [21]. 

### 2.2. Phytochemistry and Antioxidant Effects

We made a reversed phase 18 (RP_18_) TLC fractionation in order to decipher compounds or combinations of compounds in a methanol Soxhlet root extract of *A. leiocarpa* that could contribute to the good antimycobacterial effects of this extract. Our TLC-fractionation resulted in 12 fractions (Figure 4), of which four (**Fr 3**, **Fr 4**, **Fr 6**, and **Fr 8**) (Figure 4) were obtained in sufficient concentrations to be used for antimycobacterial testing and the phytochemical analysis. Our HPLC-DAD and UHPLC/QTOF-MS results of the qualitative composition of the separated TLC fractions revealed that the fractions **Fr 3**, **Fr 4**, and **Fr 6** contained ellagic acid derivatives as their main compounds (Table 3 and Figure 5B). The HPLC-DAD retention times, [M-H]^-^ data, and (UV_λ_) absorption spectra of compounds characterized from the TLC fractions of the root methanol extract of *A. leiocarpa* are shown in Table 3. Fraction 3 (**Fr 3**), the most non-polar part of the crude methanol root extract, appeared as a fluorescent blue spot on the TLC plate at 366 nm (Figure 4). Fraction 3 contained six compounds altogether, all ellagic acid derivatives, and three of these compounds—(**18**) di-methyl ellagic acid xyloside (C_21_H_18_O_12_, tR 23.47 min, [M-H]^−^ at m/z 461.0766), (**21**) di-methyl-ellagic acid (C_16_H_10_O_8_, tR 28.04 min, [M-H]^−^ at m/z 329.0318), and (**24**) an acetylated ellagic acid derivative (tR 35.08, [M-H]^−^ at m/z 343.0477)—could be characterized from this fraction (Table 3). Di-methyl ellagic acid was also found in fractions **Fr 4** and **Fr 6**, and it was the main compound both in **Fr 4** and **Fr 3** (Table 3 and Figure 4). Moreover, di-methyl ellagic acid xyloside was also present in **Fr 4**. Additionally, in fraction **Fr 4**, (**2**) protocatechuic acid (syn. dihydroxybenzoic acid, C_7_H_6_O_4_, tR 3.69 min, [M-H]^−^ at m/z 153.0196) was present, although it was assumed that this compound would mainly appear in the most polar TLC fractions (Table 3). In addition, in each TLC fraction, there were a number of ellagic acid derivatives, for which only the UV_λ_ maxima absorption spectra from HPLC-DAD could be obtained (Table 3). 

Five unknown ellagitannins (**4b**), (**19**), (**20**), (**22a**) and (**22b**) were present in the crude methanol extract of the roots of *A. leiocarpa* at tR (HPLC-DAD) 8.99, 26.10, 27.68, 31.24, and 31.58 min, respectively (Figure 5A). These ellagitannins were not found in the TLC fractions. However, a small concentration of an unknown ellagitannin (HPLC-DAD peak area 2.9%) was detected in **Fr 6** at tR 23.64 min (Table 3 and Figure 5B). 

As shown in Figure 4I,II and Table 3, we made a qualitative analysis of the antioxidant capacity of compounds in the TLC fractions of *A. leiocarpa* roots by spraying TLC plates containing these fractions with a DPPH reagent. Our results showed that all TLC fractions contained compounds with good antioxidant effects, but the strongest effects were shown by fractions **3**, **4**, and **6**. 

## 3. Discussion

Our results demonstrated that polyphenol-rich extracts and fractions of the root, leaf. and stem bark of *Anogeissus leiocarpa* possess good antimycobacterial effects against *M. smegmatis*. Surprisingly, although there have been a number of publications dealing with the antibacterial effects and phytochemistry of *A. leiocarpa*, no research has previously been done on the antimycobacterial effects of *A. leiocarpa* extracts against *M. smegmatis* or *M. tuberculosis.* Thus, our results now prompt more research on antimycobacterial fractions and compounds in this plant and related species. In the following parts of the discussion section, we discuss the most potent extracts and fractions, the impact of their extraction yields and antioxidative effects on their antimycobacterial activities, and the compounds in these extracts and fractions that are likely to contribute to their antimycobacterial effects. 

### 3.1. Extraction Yields, Total Antimycobacterial Activity and Antioxidative Effects 

The % yield of extraction in relation to the MIC gives an important indication for the total antimicrobial activity of a plant extract. Eloff, 2000 [37] presented the total activity of a plant extract as the mass extracted from the plant material (in mg/1000 mg starting material) divided by the MIC value (in mg/mL) of this extract. The total activity is an indication of the volume to which 1 g of a plant extract can be diluted without losing its antibacterial effect, and it is expressed as mL/g. In respect to this, the extraction yields resulting from our extractions when using different solvents were taken into account for *A. leiocarpa* in order to estimate the total activities of the extracts (Table 2). This information could be important to improve the use of *A. leiocarpa* as traditional medicine for the treatment of TB. In general, our results indicated that the most polar extracts, such as methanol and water extracts, gave better extraction yields when compared to the non-polar solvents (Figure 6). The extraction yields of the decoctions and macerations of *A. leiocarpa* were, however, quite low in general; 11.9% for a root bark decoction, 7.1% for a maceration of the leaves, and 4.7% for a maceration of the stem bark (Figure 5). Therefore, the use of water as extractant is maybe not optimal while preparing traditional remedies of *A. leiocarpa*. Our results suggested that ethanol extracts could be an option to macerations and decoctions to extract an optimal concentration of antimycobacterial compounds. 

Moreover, in agreement with Orlando et al. [26], we found that the extraction yields considerably varied between different plant parts, such that Soxhlet extraction with methanol resulted in a low extraction yield for the roots (10.8%) compared to the stem bark (47.1%) and leaves (50.8%) (Figure 6). Thus, the total antimycobacterial activity for the root Soxhlet methanol extract was calculated to be 43.2 mL/g (108 mg extracted from 1 g plant material divided by the MIC in mg/mL, 2.5 mg/mL). For the leaves, the total activity was 101.6 mL/g, even though the MIC value for the leaf Soxhlet methanol extract was as high as 5000 µg/mL. Thus, despite of the lower MIC value of the root methanol extract compared to the leaf methanol extract, the total activity of the leaf Soxhlet extract was better (Table 2). This also implies that leaf extracts in alcohol of *A. leiocarpa* could be good sources for antimycobacterial compounds, since the extraction yields with alcohol are high. 

Small extraction yields could limit the use of some extracts of *A. leiocarpa* (and plants in general) as sources of antimycobacterial compounds. Moreover, the active compounds can sometimes be present in very small concentrations, which could be a hinderance when trying to elucidate antimycobacterial compound structures [38]. 

According to our qualitative DPPH-TLC assay of thin-layer chromatography fractions obtained from a methanol root extract of *A. leiocarpa*, some of the fractions, such as **Fr 3**, **Fr 4**, and **Fr 6**, gave strong antioxidative effects (Figure 4). These antioxidant effects could have positive impacts on the antimycobacterial effects of the mentioned TLC fractions. Our results are in accordance with previous research that *A. leiocarpa* contains a number of polyphenolic compounds with good antioxidant effects [32]. 

### 3.2. Ellagic Acid Derivatives and Ellagitannins in A. Leiocarpa and Their Suggested Impact on Its Antimycobacterial Effects 

In our screenings, the lowest MIC value (625 µg/mL) against *M. smegmatis* was obtained with an ethyl acetate extract of the roots of *A. leiocarpa*. The good antimycobacterial effects of this extract could have been due to ellagic acid derivatives and ellagitannins in this extract. In addition, we observed that the antimycobacterial effects of a methanol extract of the root (MIC 2500 µg/mL) was improved when it was fractionated using preparative reversed phase RP_18_-TLC and methanol:water:acetic acid (50:50:1) as eluent. When compared to the crude extract, fractions **3** and **4**, enriched with ellagic acid derivatives, gave lower MIC values of 1500 and 1000 µg/mL, respectively (Table 2). Di-methyl ellagic acid xyloside (Figure 3D) is especially suggested to strongly contribute to the antimycobacterial effects of both fractions (Figure 5 and Table 3), since di-methyl ellagic acid xyloside inhibited the growth of *Mycobacterium* intracellulare and showed an outstanding MIC value of 4.88 μg/mL against *M. smegmatis* and *M. tuberculosis* MTCS_2_ [39]. The differences in MIC values, when compared to our study (4.88 µg/mL versus 1000–1500 µg/mL in our study) could have been due to the fact that our thin layer fractions contained a small total concentration of di-methyl ellagic acid xyloside (4.9% and 9% peak area in HPLC-DAD chromatograms, in fraction **3** and **4**, respectively) and, in addition, contained a mixture of unknown ellagic acid derivatives. Moreover, Kuete et al. [39] used a different strain of *M. smegmatis*. Besides, di-methyl ellagic acid xyloside has also been identified from the stem bark of another species of *Anogeissus*, *A. latifolia* [40].

The xylose sugar part of di-methyl ellagic acid xyloside is thought to be important for its antimycobacterial activity [41], and accordingly, the aglycone di-methyl ellagic acid was found to be less active. In agreement with this, our screening result of ellagic acid against *M. smegmatis* (MIC 500 µg/mL) showed that pure ellagic acid is not very active, and it needs to have a sugar molecule or other side groups attached to increase its activity. Thus, ellagic acid derivatives, and especially ellagic acid glycosides from *Anogeissus leiocarpa* (Figure 3) and related species, might have a significant value as scaffolds for new anti-TB drugs. 

Another ellagic acid derivative isolated from *A. leiocarpa*, 3,3,4′-tri-*0*-methyl-flavellagic acid, was found to inhibit the growth of *S. aureus*, *E. coli,* and *P. aeruginosa* [42] but was not tested for its growth inhibition against mycobacterial strains. Thus, *Anogeissus* spp. are good sources for antibacterial ellagic acid derivatives with antibacterial properties.

Though some ellagic acid derivatives have been found to give strong antimycobacterial effects, not much is known about their mechanisms of action on *Mycobacterium* spp. It has been suggested that ellagic acid derivatives interfere with mycobacterial cell wall synthesis [43]. In accordance with this suggestion, a study revealed that pteleoellagic acid had a high in silico docking score to MabA, a protein involved in the fatty acid elongation complex II (FAS II) system in *M. tuberculosis* [44]. Additionally, di-methyl ellagic acid xyloside inhibits arabinogalactan synthesis in the mycobacterial cell wall [43,44]. 

*A. leiocarpa* roots and stem bark are rich source of ellagitannins (ET). In this investigation, we found that an ethyl acetate extract of *A. leiocarpas* roots gave the lowest MIC of 625 µg/mL against *M. smegmatis* (Table 2). The good antimycobacterial effects of this extract could partly be attributed to its ellagitannins. Moreover, we found that an ethyl acetate extract of the stem bark, enriched with ellagitannins, gave fairly good antimycobacterial activity. In our previous investigation, we tentatively characterized ten unknown ellagitannins in this stem bark ethyl acetate extract of *A. leiocarpa* based on HPLC-UV-DAD data (UV_λ_ maxima and retention times) [18]. Moreover, previously reported ellagitannins in the stem bark and leaves of *A. leiocarpa* include a corilagin sanguiin H-4, punicalagin, castalagin (1-epi-vescalagin), casuarinin, chebulagic acid, and punicacortein (Figure 7) [26,31]. These mentioned ellagitannins may play a significant role for the in vitro antimycobacterial effect of the stem bark ethyl acetate extract of *A. leiocarpa* that we present in this paper. In this respect, we found that corilagin expressed a mild growth inhibitory effect against *M. smegmatis* with an MIC value of 1000 µg/mL (Table 2). Some other ellagitannins are also known to possess in vitro growth inhibitory effects against mycobacterial strains. For example, Asres et al., 2001 [45] found that punicalagin isolated from the stem bark of *Combretum molle* completely inhibited the growth of *M. tuberculosis* ATCC 27294 at a concentration of 1000 µg/mL, and Diop et al., 2019 [46] found that fractions enriched with the ellagitannin isomers punicalagin α and β effectively inhibited the growth of *Mycobacterium marinum*, whereas a fraction enriched with punicacortein D showed a weaker effect.

However, ETs are mostly too large to be absorbed as such in vivo, and in the human body, they are mainly present as their degradation products of urolithins [47]. Thus, the role of ETs in anti-TB traditional remedies might mostly be due to their metabolically available products, the urolithins [46,48]. Since we also found that the same ellagitannins were present in a water extract of the stem bark of *A. leiocarpa* (HPLC-DAD result not published), this could justify the traditional use of water extracts (decoctions and macerations) of *A. leiocarpa* for the treatment of tuberculosis and its symptoms, such as (bloody and prolonged) cough, assuming that the urolithins and ellagic acid and its derivatives, resulting from the degradation of the ETs would be antimycobacterially active or that some metabolically unchanged smaller molecular ETs would be active constituents in the extract. Accordingly, in their investigation on antimycobacterial compounds of *Combretum aculeatum*, Diop et al., 2019 [46] suggested that the effects of the traditional decoction of this plant could mostly be due to ellagic acid and urolithins. 

In vitro investigations on the mechanism of action of ETs on bacteria have indicated that the presence of galloyl groups and the presence and number of dehydro-hexahydroxydiphenoyl ester (D-HHDP) and hexahydroxydiphenoyl ester groups (HHDP) in ellagitannins influence their antimicrobial effects [9,49,50].

### 3.3. Flavonoids in A. Leiocarpa and Their Suggested Impact on Its Antimycobacterial Effects 

Our results indicated that a methanol Soxhlet extract of the root of *A. leiocarpa* have antimycobacterial effects. Flavonoids could have partly been behind these effects, since in our previous investigation, we found that this extract is rich in flavonoids, such as aromadendrin (syn. dihydrokaempferol), ampelopsin (syn. dihydromyricetin), methyl-taxifolin (syn. dihydroisorhamnetin), and taxifolin (syn. dihydroquercetin) [18]. Moreover, we have now found some additional flavonoids in this root extract, as shown in Figure 5A, at HPLC-UV-DAD Rt 4.74, 8.64, 14.37, and 14.38 min. The molecular masses for these flavonoids remain to be elucidated. 

Previous research has demonstrated that methanol and water extracts of the stem bark and leaf of *Anogeissus leiocarpa* are rich in flavonoids such as luteolin, apigenin, quercitrin, isorhamnetin, isoquercetin, quercetin-3-*O*-glucoside, luteolin-7-*O*-glucoside, kaempferol, rutin, quercetin-*O*-galloylhexoside, myricetin-*O*-hexoside, apigenin-6-*C*-glucoside, reinutrin, avicularin, myricetin, guaijaverin, eriodictyol, narcissin, naringenin, pinocembrin, quercetin, catechin, and vitexin [26,32,42]. In addition the dimer, procyanidin B-2 was found to be present in an ethyl acetate extract of the leaves [32]. 

Some flavonoids occurring in *A. leiocarpa* have been tested for their antimycobacterial effects. For example, taxifolin was found to give outstanding growth inhibitory effects against *M. tuberculosis* H37Rv with an MIC value of ≤ 12.5 μg/mL [51,52]. Thus, the taxifolin that we found in high concentrations in a methanolic root extract of *A. leiocarpa* could be an important contributor to the growth inhibitory effects of this extract against *M. smegmatis*. Additionally, isorhamnetin showed potent growth inhibitory effects against multi-drug and extensively drug-resistant clinical strains of *M. tuberculosis* H37 Rv [53]. Moreover, quercetin-3-*O*-glucoside gave an MIC of 150 µg/mL against *M. tuberculosis* H37 Rv [54]. Most of the flavonoids that have been investigated for their antimycobacterial effects have been found to possess moderate to weak growth inhibitory effects. For example, quercetin showed MIC values of 200 µg/mL against *M. tuberculosis* H37 Rv [55], and luteolin gave an MIC value of 699 µM against *M. tuberculosis* [56]. Thus, our results on the weak growth inhibitory effects of apigenin and quercetin against *M. smegmatis* (both flavonoids gave MIC values of 250 µg/mL in our test system) are in line with these previous findings. 

Many authors’ results have suggested that the (most of the) flavonoids in plant extracts (including extracts of *A. leiocarpa*) act in concert with other compounds present in the extracts, rather than functioning as the principal antimycobacterial compounds alone. Indeed, it has been found that fractions containing combinations of flavonoids, such as a fraction enriched in quercetin and guercitrin isolated from a Brazilian medicinal plant *Scutia buxifolia* (Rhamnaceae), gave promising growth inhibitory effects against *M. smegmatis* (MIC 78 µg/mL) [57]. Moreover, some flavonoids have been found to act synergistically with conventional antimycobacterial drugs. For example, a trihydroxylated methoxychalcone isolated from *Galenia africana* (Aizoaceae) was found to have synergistic effects in combinations with isoniazid against *M. tuberculosis* [58]. In addition, some flavonoids have been found to possess powerful antimycobacterial effects when tested alone, such as isobachalcone (MIC 2.2 µg/mL against *M tuberculosis*) isolated from the West-African plant *Dorstenia barteri* (Moraceae) [59]. Thus, in depth efforts should be made to investigate the flavonoid components in *Anogeissus* spp. and to isolate these flavonoids in order to investigate their antimycobacterial effects alone and in combinations with other flavonoids and with conventional anti-TB drugs, such as rifampicin and isoniazid.

Some structure–activity relations and mechanisms of action have been studied regarding the antimycobacterial effects of flavonoids. It has been found that the antimycobacterial potential of flavonoids is mainly dependent on the presence of a heterocyclic ring, the presence and position of functional groups (-OH, -OMe) and a carbon composition of C_6_-C_3_-C_6_ in their molecular structure [60]. Additionally, it has been found that the flavonoid structure is suitable for the development of *M. tuberculosis* proteasome inhibitors **[61,62].** DNA-gyrase in *M. tuberculosis* is an important drug target, and taxifolin and quercetin are able to bind to this protein [52,63]. Moreover, flavonoids have lipid peroxidation activity and antioxidant activity, which could also affect their antimycobacterial potential [64]. Additionally, some flavonoids, such as butein and isoliquiritigenin have been found to inhibit fatty acid and mycolic acid synthesis and thus to affect the formation of the mycobacterial cell membrane and cell wall [3].

## 4. Materials and Methods 

### 4.1. Plant Material 

The plant material of *Anogeissus leiocarpa* was collected from a natural woodland forest in the Blue Nile region in southeastern Sudan (Figure 1). The plants were identified and authenticated at the Faculty of Forestry, University of Khartoum, Sudan, and voucher specimen and voucher numbers were deposited at the University of Khartoum, Sudan. The stem, leaves, fruits, and roots of the plant specimens were air-dried in the shade. Stems and roots were manually debarked to separate the stem bark from the wood and the root bark from the root. The dried plant parts were manually chipped to small pieces and ground using a grinder machine. 

### 4.2. Extraction 

#### 4.2.1. Cold and Hot Methanol Extraction

For the cold methanol extraction, 20 g of the plant material was added to 400 mL of methanol and stirred overnight in a magnetic stirrer (RCT basic digital). The resulting extracts were centrifuged at 2500 rpm for 10 min (Eppendorf AG centrifuge 5810 R, Germany), whereafter the supernatants were freeze-dried in a lyophilizer (HETO LyoPro 3000, Denmark) for three days. 

For the hot methanol extraction using the Soxhlet technique, 800 mL of methanol was added to 20–100 g of powder from various plant parts. Extraction was performed for 5 h. A rotary evaporator (Heidolph VV2000) was used to evaporate the solvent of the extracts, after which the extracts were freeze-dried for 1–2 days.

The extracts obtained from these methods were dissolved in methanol (50 mg/mL) for antimycobacterial screening. Additionally, the methanolic Soxhlet extracts at 5 mg/mL (in MeOH: H_2_O, 1:1) were subjected to further UHPLC/QTOF-MS phytochemical analysis. 

#### 4.2.2. Macerations and Hot Water Decoctions

Cold and hot water extracts were prepared in accordance with African traditional medicinal knowledge. For the macerations, 500 mL of distilled water was added to 20 g of the plant powder and extracted for 24 h using a magnetic stirrer (RCT basic digital). For the decoctions, 500 mL of water were added to the plant powder and left to boil for 5 min. 

Extracts obtained from both methods were transferred into Eppendorf centrifuge tubes (volume 50 mL, Germany) and centrifuged at 3000 rpm (Eppendorf AG centrifuge 5810 R, Germany) for 15 min. The resulting supernatants were collected and filtered using a vacuum pump filtration technique and Whatman filter paper (⌀ 150 mm, Germany). The filtered extracts were placed in −20 °C for 24 h, whereafter they were lyophilized to dryness for 1–3 days using a (HETO LyoPro 3000, Denmark) lyophilizer. 

#### 4.2.3. Sequential Extraction and Solvent Partition 

In brief, sequential extraction was performed according to the method described by Salih et al. 2017a and 2017b [18,65]. A total of 100 g of the plant materials of *A. leiocarpa* were subjected to a sequential extraction method using solvents with increasing polarities. First, the plant material was extracted with hexane (1000 mL), followed by dichloromethane or chloroform (1200 mL) and acetone (900 mL) in the case of the root and root bark. Following this sequential extraction, liquid–liquid fractionation was used. In this method, ethyl acetate was added to extracts that contained 80% methanol, and separation was performed in a separation funnel (a Nalgene® FEP). All extracts were dried in the rotary evaporator, whereafter they were freeze-dried in a lyophilizer. The obtained extracts were subjected to antimycobacterial testing.

The percentage extraction yields for the extracts were calculated as the ratio of the extract dry weight to the dry weight of the plant material used for the extraction according to the below formula [66]: $$\mathrm{Percentage}\;\mathrm{extraction}\;\mathrm{yield}\;\left(\%\right)=\left(\frac{\mathrm{weight}\;\mathrm{of}\;\mathrm{the}\;\mathrm{dry}\;\mathrm{extract}}{\mathrm{weight}\;\mathrm{of}\;\mathrm{dry}\;\mathrm{plant}\;\mathrm{material}\;\mathrm{before}\;\mathrm{extraction}}\right)\;\times \;100$$

### 4.3. Phytochemical Analysis

#### 4.3.1. Thin Layer Chromatography and Antioxidant Analysis Using The DPPH-Reagent

A root extract of *A. leiocarpa*, extracted using hot methanol Soxhlet extraction, was subjected to preparative and analytic reversed phase TLC and antioxidant analysis to isolate fractions enriched with ellagic acid derivatives and to detect the antioxidant compounds in these fractions. Reversed phase silica gel (RP_18_ F 254 s, Merck, Germany) plated glass plates (20 × 20 cm) were used for the separations, and aluminum backed plates with the same solid phase were used for analytical TLC. A volume of 10 μL of the standard compounds, ellagic acid, corilagin, and gallic acid (Sigma-Aldrich, 1 mg/mL) was applied on the thin layer plates. Additionally, 20–30 µL of extracts in methanol (50 mg/mL) were applied in 5–10 µL volumes at a time (using a glass Pasteur pipette, P 4518-5X, Accupipette ^TM^ Pipets/ DADE and DURAN®, ring Caps, Germany) at the baseline of the thin layer plate and allowed to dry between the applications. Compound and pure compound applications were repeated 2–3 times. The eluent for the TLC separations consisted of methanol:water:orthophosphoric acid (50 mL:50 mL:1000 µL). The plates were developed until the solvent front reached 12.7 cm (application baseline at 1.5 cm). The dried plates were examined under a UV-detector (Camaq Reprostar 3 TLC Visualizer) to identify the fluorescent and quenching bands at 366 and 254 nm, respectively. The retardation factor (R_f_) of the fractions was measured using the below formula as the relation of the distance moved by the fractions to the distance moved by the eluent front:$$\mathrm{Rf}=\frac{\mathrm{Distance}\;\mathrm{moved}\;\mathrm{by}\;\mathrm{the}\;\mathrm{fractions}}{\mathrm{Distance}\;\mathrm{moved}\;\mathrm{by}\;\mathrm{the}\;\mathrm{solvent}}$$

For the preparative thin-layer fractionation, the fractions were scraped from 15 plates, and bands showing the same Rf value were combined into the same glass tube. Methanol (2–5 mL) was added to the glass tubes with the fractions in order to extract the compound (-s) from the silica gel. After this extraction step, the glass tubes were centrifuged at 5000 rpm for 10 minutes (Mini Spin^®^ plus, Eppendorf, Germany) and the supernatants were collected. In order to obtain the dry weights of the obtained fractions, the methanol was evaporated using a heating centrifuge (Concentrator Plus, Eppendorf), whereafter the separated fractions were weighed and dissolved in methanol:water (1:1) for HPLC-DAD- and UHPLC/QTOF MS-analysis. 

The DPPH (2,2-diphenyl-1-picrylhydrazyl, C_18_H_12_N_5_O_6_, Sigma-Aldrich) reagent (0.2% w/v in methanol) was used for the qualitative evaluation of the oxidative reaction of the separated spots on the thin layer plates in visual light. The antioxidant compounds turned their color from violet to yellowish or colorless due to reactions and neutralization with the DPPH of the various functional groups in the detected compounds. The separated spots were compared with the standard compounds corilagin, ellagic acid, and gallic acid (Sigma-Aldrich). 

#### 4.3.2. HPLC-UV/DAD Method

Analysis was performed using the Agilent Chemstation HPLC-UV/DAD system (Waters Corp., Milford, USA) and employing the method described by Salih et al. 2017a [18] and Salih 2019 [9]. The HPLC was connected with a controller, a water pump (600 E), and a diode-array detector UV (a 991 PDA). The separation was performed using an analytical Hypersil reversed phase C-18 column (length: 10 mm; ID: 2 mm). The eluents of the mobile phase consisted of solvent A, an aqueous solution that consisted of 1.5% tetrahydrofuran and 0.2% of orthophosphoric acid, and solvent B, which was 100% methanol. Gradient elution was used as described by Salih et al., 2017a [18], and the flow rate and injection volume were 2 mL/min and 10 μl (2 mg/mL in 50% MeOH), respectively. The wavelengths were set at 220, 270, 280, 320, and 360 nm, and the UV_λ_ absorption maxima spectra of the identified compounds were recorded. The compounds were compared with a natural compounds library available in the computer Agilent Chemstation library. Moreover, the compounds were compared with data reported in literature [9,18,67,68].

#### 4.3.3. UHPLC/Q-TOF MS Method 

The methods described by Taulavuori et al. 2013 [69], Fyhrquist et al., 2014a [34], and Salih et al. 2017a [18] and Salih 2019 [9] were used to characterize the compounds of interest and to obtain the molecular ions in negative mode as [M-H]^-^ values. The UHPLC/Q-TOF MS apparatus was equipped with UHPLC-DAD for the compound separations (Model 1200 Agilent Technologies)-JETSTREAM/QTOFMS (Model 6340 Agilent Technologies) and a reversed phase column (2.1 × 60 mm, 1.7 µm C_18_ column, Agilent technologies). The mobile phase included solvent A, 1.5% of tetrahydrofuran and 0.25% of acetic acid in ionized water, and solvent B was 100% methanol. The UHPLC-runs were made using the following gradient: from 0 to 1.5 min, 0% B; from 1.5 to 3 min, 0–15% B; from 3 to 6 min, 10–30% B; from 6 to 12 min, 30–50% B; from 12 to 20 min, 50–100% B; and from 20 to 22 min, 100–0% B. All molecular ions were acquired in the negative ion mode with the mass range from 100 to 2000 m/z. The acquired ions were compared with the literature [67]. 

### 4.4. Antimycobacterial Activity Tests

#### 4.4.1. Agar Diffusion Method

The methods described by Fyhrquist et al. 2014 b [70] and Salih et al. 2018 [71] were used. *Mycobacterium smegmatis* ATCC 14468 was grown for five days at +37 °C on agar slants containing a Löwenstein–Jensen agar medium (Becton–Dickinson and Company, USA). For the test, petri dishes (Ø 14 cm, Bibby Sterilin, UK) that were filled with a base layer of 25 mL agar (Antibiotic medium Number 2, Difco, Molesey, UK) and 25 mL Middlebrook 7H10 agar (Difco) supplemented with oleic albumin dextrose catalase supplement (OADC, Difco) as a top layer were used. 200 μL of *Mycobacterium* suspension in sodium chloride (0.9% w/v) containing approximately 1.0 × 10^8^ CFU/mL was aseptically applied to the petri dish. After this inoculation, filter paper disks (Ø 12.7 mm, Schleicher and Schuell) loaded with 200 μL of the extracts (50 mg/mL) and the positive control, rifampicin (10 mg/mL, Sigma-Aldrich) were aseptically and equidistantly placed on the petri dishes. Then, 200 µL of the solvents that were used to dissolve the plant extracts, methanol and hexane, were used as negative controls. Prior to incubation, all petri dishes containing extracts and positive and negative controls were left in the cold room (+4 °C) for two hours. Incubation was at +37 °C for five days. Experiments were performed in triplicates and the diameters of the IZD were measured in millimeters. The results were expressed as the mean IZD of the triplicates ± standard error of means (SEM). Moreover, the activity index (AI) of the various extracts was evaluated as the relation of the IZD of the extracts compared to the IZD of rifampicin, as shown in the equation below [70].
$$\mathrm{AI}\;\left(\mathrm{Activity}\;\mathrm{Index}\right)=\frac{\mathrm{Inhibition}\;\mathrm{zone}\;\mathrm{of}\;\mathrm{the}\;\mathrm{plant}\;\mathrm{extract}}{\mathrm{Inhibition}\;\mathrm{zone}\;\mathrm{of}\;\mathrm{rifampicin}}$$

Agar diffusion was used for the measurement of approximate MIC values for some of the extracts due to the excess turbidity resulting from precipitation of the compounds in these extracts (such as some methanol and hexane extracts). Thus, the MIC values for these extracts could not be measured with the microplate method. In brief, for these experiments, 200 µL of two-fold dilutions of the extracts were prepared (5 mg/mL–39 µg/mL dilutions) and pipetted on filter papers to determine the MIC value for these extracts. The lowest concentration that showed a small but still visible inhibition zone (≤1 mm) was considered as the minimum inhibitory concentration. 

#### 4.4.2. Turbidimetric Microplate Method

Using the methods described in Fyhrquist et al. 2014 b [70] and Salih et al. 2018 [71], extracts, pure compounds, positive and negative controls, and fractions obtained from the thin layer chromatography were subjected to a turbidimetric microplate test to obtain the minimum inhibitory concentration. Two-fold dilutions of the plant extracts were made in methanol from 5 to 39 µg/mL. The preparative TLC fractions (**Fr 3**, **4**, **6**, and **8**) from a methanolic root extract of *A. leiocarpa* were diluted in methanol to achieve concentrations of 1500, 1000, 2000, and 3000 µg/mL, respectively, and two-fold dilutions were made from these stock solutions until 3.91 µg/mL was reached. Rifampicin was diluted from 1000 to 1.953 µg/mL). The standard compounds, gallic acid (G-7384, Sigma-Aldrich, Dermstadt, Germany), ellagic acid (Sigma-Aldrich), corilagin (Sigma-Aldrich), quercetin (Merk Art. 7546, Darmstadt, Germany), and apigenin (Etrasynthese 69730 Genay, France) were used as reference compounds that are known to occur in *A. leiocarpa*. These reference compounds were prepared as two-fold dilutions from 1000 to 0.030 μg/mL.

Before the microplate test, a few colonies of *Mycobacterium smegmatis* ATCC 14468 were transferred to 15 mL of the Dubos broth (Difco) and incubated for three days at +37 °C and 200 rpm. For the test, 2 mL of the suspension were pipetted into a glass tube; 1 mL was then pipetted into a disposable UV Cuvette (BrandTech^®^, Essex, NJ, USA) to measure the turbidity at 625 nm using a UV–visible spectrophotometer (Pharmacia LKB-Biochrom 4060). The remaining suspension in the glass tube was then diluted to reach an absorbance of 0.1 (containing approximately 1 × 10^8^ CFU/ mL). This suspension was diluted further according to the guidelines of the Clinical and Laboratory Standards Institute 2013 [72], who recommend an inoculum of 2.5 × 10^5^ CFU/mL for *Mycobacterium smegmatis* MIC testing. A volume of 100 µl of suspension containing 5 × 10^5^ CFU/mL was added to the 96-well microplate wells (Nunc, Nunclon, Roskilde, Denmark) containing 100 µL of extracts, TLC fractions, pure compounds, or rifampicin, thus resulting in a final inoculum of 2.5 × 10^5^ CFU/mL. A total of 100 µL of the growth control (bacterial suspension containing 5 × 10^5^ CFU/mL) and 100 µL of broth were added to the growth control wells to reach the same CFU/mL as in the test wells (a final inoculum of 2.5 × 10^5^ CFU/mL). Moreover, solvent controls (methanol and hexane) were added to the wells in the same maximum percentages as in the wells containing the highest concentrations of extracts (that meant a maximum volume percentage of 5% for the solvents). The solvents (at max. 5% volume of the suspension) were found to not be toxic to *M. smegmatis*. The microplates were incubated at +37 ˚C for four days, whereafter the turbidity of the wells was measured at 620 nm using a Victor 1420 (Wallac, Turku, Finland) spectrophotometer.

All assays were done in triplicate, and as shown in the equation below, the results are expressed as the mean percentage of growth inhibition of triplicates compared to the growth of the growth control (= 100% growth). Sample controls (SC, containing only plant extract and broth, without bacterium) were used to subtract the turbidity resulting from the plant extracts (the turbidity of test well with plant extract and bacterium minus turbidity of sample control containing the same plant extract but no bacteria). The smallest concentration that inhibited ≥90% of the mycobacterial growth was considered as the MIC value [71].

Formula (1 and 2) used for the calculation of growth inhibition:(1)$$\%\;\mathrm{bacterial}\;\mathrm{growth}\;=\left[\frac{\left(\;\bar{x}\mathrm{GT}\;A620\;-\;\bar{x}\mathrm{SC}\;A620\right)}{\;\bar{x}\mathrm{GC}\;A620}\;\times \;100\right]$$
(2)$$\%\;\mathrm{inhibition}\;\mathrm{of}\;\mathrm{growth}=100\;(\%\;\mathrm{growth}\;\mathrm{of}\;\mathrm{the}\;\mathrm{growth}\;\mathrm{control})\;-\;\left[\frac{\left(\;\bar{x}\mathrm{GT}\;A620\;-\;\bar{x}\mathrm{SC}\;A620\right)}{\;\bar{x}\mathrm{GC}\;A620}\;\times \;100\right]$$
where GT A_620_ is the turbidity of the test well at 620 nm (containing plant samples, pure compounds or antibiotics and microbial cells), SC A_620_ is the sample control (consisting of the plant samples, the compounds or antibiotics alone, without microbial cells), and GC A_620_ is the turbidity of the growth control at 620 nm (containing only bacterial cells). $\bar{x}$
is the average of the triplicates.

In addition, the total activity of a plant extract was calculated as the mass extracted from the plant material (in mg/1000 mg starting material) divided by the MIC (in mg/mL) value of this extract.

## 5. Conclusions

The results from this investigation indicated that extracts of the African medicinal plant *Anogeissus leiocarpa* contains compounds that might have use for the treatment of tuberculosis. Our results could justify the traditional oral application of utilizing water extracts and decoctions of *A. leiocarpa* roots, leaves, and stem bark to treat cough related to TB. Moreover, our results indicate that ethanol extracts could be preferred for traditional use, since alcohol extracts more compounds than water. 

To the best of our knowledge, this is the first time extracts and fractions of *A. leiocarpa* have been tested for their antimycobacterial effects against *M. smegmatis*. Our RP_18_-TLC fractionation of *A. leiocarpa* roots resulted in the separation of some ellagic acid enriched fractions that were more active than the crude methanol extract. Of the active compounds in these fractions, di-methyl ellagic acid and di-methyl ellagic acid xyloside were enriched in some of the fractions, and these compounds have been found to possess promising antimycobacterial effects. Thus, *A. leiocarpa* could be a good source for these antimycobacterial ellagic acid derivatives. 

Ellagic acid-based glycosides are difficult to synthesize due to the *O*-glycosylation step, and thus the possibility to use plants (and their renewable organs such as leaves) as sources for EA glycosides should be investigated. 

Our results indicate that in addition to its ellagic acid derivatives, *A. leiocarpa* contains a high variety of compounds with potential antimycobacterial activity such as flavonoids (especially taxifolin) and stilbenes (pinosylvin and 4′-methylpinosylvin). Moreover, a number of ellagitannins have been characterized in *A. leiocarpa* that could have good antimycobacterial effects in vivo via their metabolically available urolithins.

Further in-depth studies should be performed on fractions and compounds isolated from *A. leiocarpa*, to investigate their individual effects on various strains on *M. tuberculosis,* as well to study the antimycobacterial potential of extracts and isolated fractions/compounds in combinations with conventional anti-TB drugs, such as rifampicin and isoniazid. 

## Figures and Tables

**Figure 1 antibiotics-09-00364-f001:**
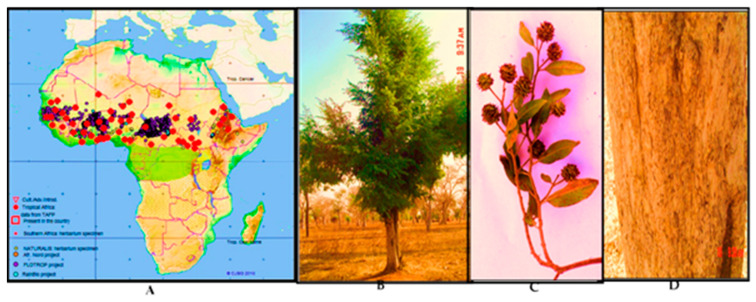
(**A**) Occurrence of *Anogeissus leiocarpus* in Africa according to various vegetation mapping projects. (TAFP), red dots; Naturalis Herbarium specimen, yellow spots; Africa Nord project, orange spots; FLOTRO project, violet dots (Source: African Plant Database); (**B**) *Anogeissus leiocarpa* growing in semi-arid and woodland savannah zone in Sudan; (**C**) leaves and flowers; (**D**) scaly bark. Photos: E. Y. A. Salih and H. H. Gibreel 2006.

**Figure 2 antibiotics-09-00364-f002:**
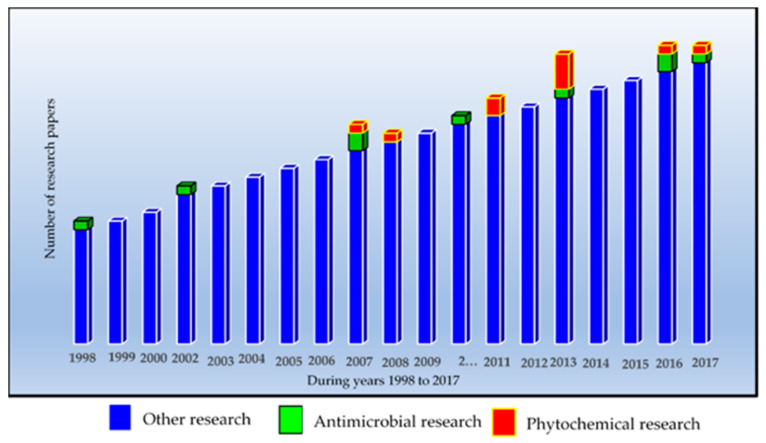
Trend in the number of research papers on *Anogeissus leiocarpa* from 1998 to 2017. The figure illustrates that most of the research papers have dealt with other research aspects than antimicrobial or phytochemical research. Thus, there is a need for more research in the mentioned research areas. Source: Scopus research data base.

**Figure 3 antibiotics-09-00364-f003:**
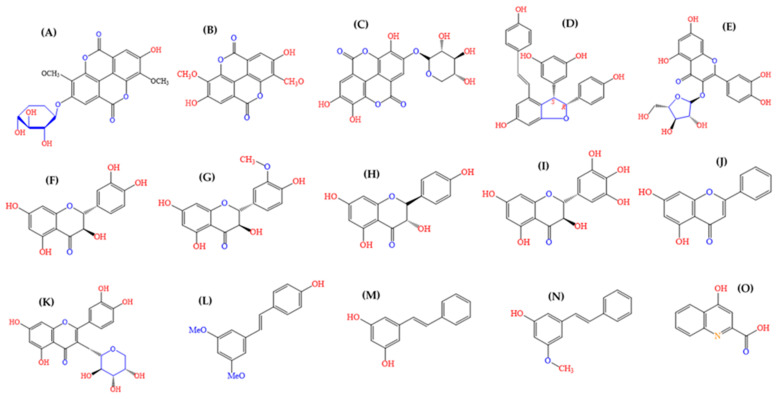
Chemical structure of ellagic acid derivatives and other chemical constituents detected earlier from *A. leiocarpa*: (**A**) dimethyl ellagic acid xylopyranoside, (**B**) dimethyl ellagic acid, (**C**) ellagic acid xylopyranoside, (**D**) epsilon-Viniferin, (**E**) avicularin, (**F**) taxifolin (dihydroquercetin), (**G**) methyl-taxifolin (dihydroisorhamnetin), (**H**) aromadendrin (dihydrokaempferol), (**I**) ampelopsin (dihydromyricetin), (**J**) pinocembrin, (**K**) guaijaverin, (**L**) pterostilbene, (**M**) pinosylvin (trans-dihydroxystilbene), (**N**) methyl pinosylvin, and (**O**) kynurenic acid [18,26,27]. Source for molecular structures: (CAS), 2020.

**Figure 4 antibiotics-09-00364-f004:**
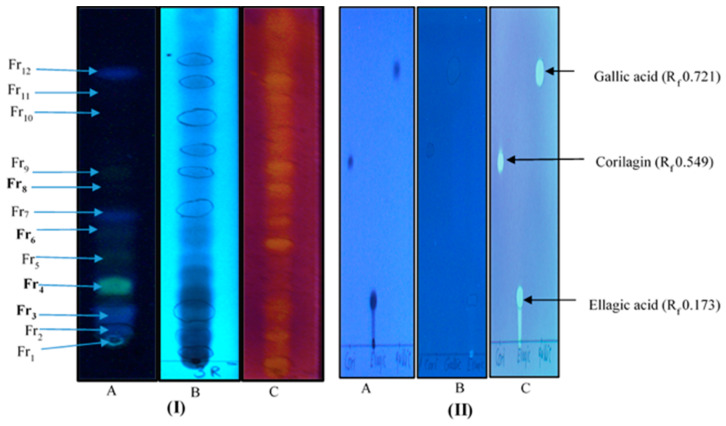
(**I**) RP_18_-TLC fractions (**Fr1**–**Fr12**) of a Soxhlet extract obtained with methanol from the root of *A. leiocarpa*. (**A**) Fractions at 366 nm, (**B**) fractions at 254 nm, and (**C**) fractions sprayed with the 2,2-diphenoyl-1picrylhydrazy (DPPH) reagent, with the yellow color indicating antioxidant activity. (**II**) Thin layer chromatography fingerprints of the standard compounds corilagin (R_f_ 0.549), ellagic acid (R_f_ 0.173), and gallic acid (R_f_ 0.721) (**A**) at 254 nm, (**B**) at 366 nm, and (**C**) after being sprayed with the DPPH reagent. Fractions marked with bold font (**Fr 3**, **Fr 4**, **Fr 6**, and **Fr 8**) were obtained in sufficient amount for antimycobacterial testing.

**Figure 5 antibiotics-09-00364-f005:**
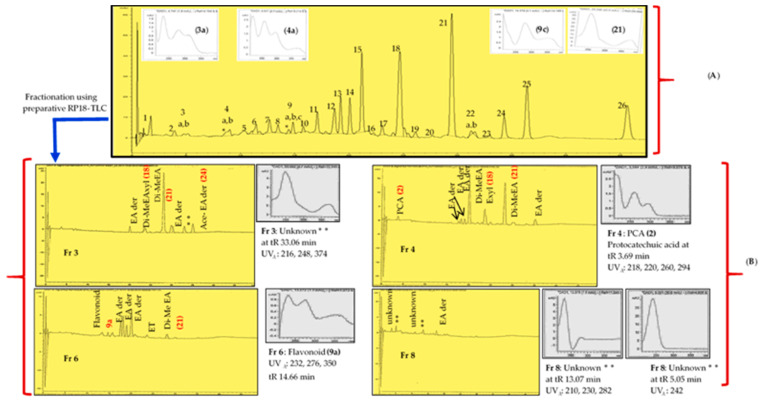
(**A**) Methanolic Soxhlet extract of the root of *A. leiocarpa*. (**1**) Gallic acid, (**2**) protocatechuic acid, (**3a**) flavonoid, (**3b**) alkaloid, (**4a**) flavonoid, (**4b**) ellagitannin, (**5**) aromadendrin, (**6**) di-galloyl-β-d-glucose, (**7**) ampelopsin, (**8**) methyltaxifolin, (**9a***) flavonoid, (**9b**) isomer of methyltaxifolin, (**9c**) flavonoid, (10) ellagic acid derivatives, (**11**) taxifolin, (**12**) ellagic acid derivatives, (**13**) ellagic acid derivatives, (**14**) ellagic acid derivative, (**15**) di-methyl ellagic acid glucoside, (**16**) pentagalloylglucose, (**17**) ellagic acid derivatives, (**18**) di-methyl ellagic acid xylopyranoside, (**19**) ellagitannin, (**20**) ellagitannins, (**21**) di-methyl ellagic acid, (**22a**) ellagitannin, (**22b**) ellagitannin, (**23**) pinosylvin, (**24**) acetylated ellagic acid derivative, (**25**) ellagic acid derivatives, and (**26**) methylpinosylvin. (**B**) TLC chromatograms of RP_18_ TLC fractions (**Fr3**–**Fr8**) obtained from the methanol Soxhlet extract (**A**); ** Unknown compounds; (EA der), ellagic acid derivatives; (Di-Me EA xyl), dimethyl ellagic acid xyloside; (ET), ellagitannin; (Ace-EA der), acetylated ellagic acid derivatives; and (PCA), protocatechuic acid. Main compounds with bold text.

**Figure 6 antibiotics-09-00364-f006:**
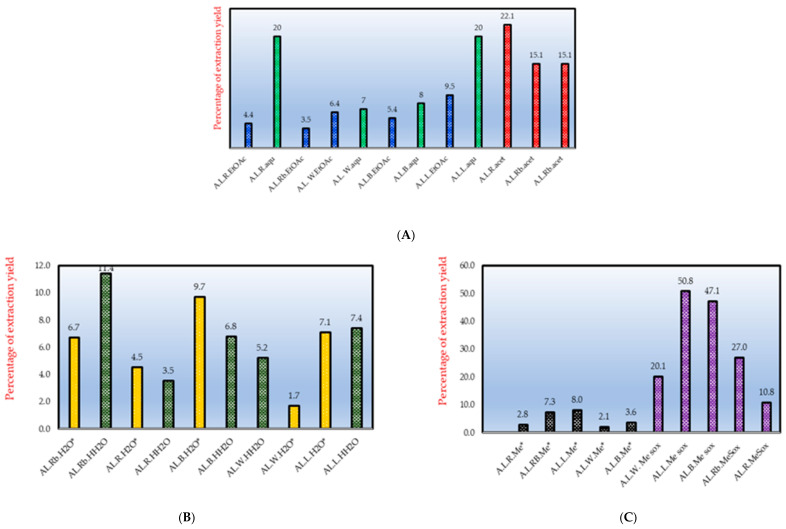
Percentage of extraction yield. (**A**) ethyl acetate (EtOAc), aqueous (aqu), and acetone (acet) in sequential extraction; (**B**) hot (HH_2_O) and cold water (H_2_O*) extraction; and (**C**) methanol extraction, Soxhlet extraction (Me sox), and cold methanol extraction (Me*); (A.L.), *Anogeissus leiocarpa*; (Rb), root bark; (R), root; (W), stem wood; (B), stem bark; and (L), leaf. Extracts obtained using the same solvent are marked with the same color or color–texture.

**Figure 7 antibiotics-09-00364-f007:**
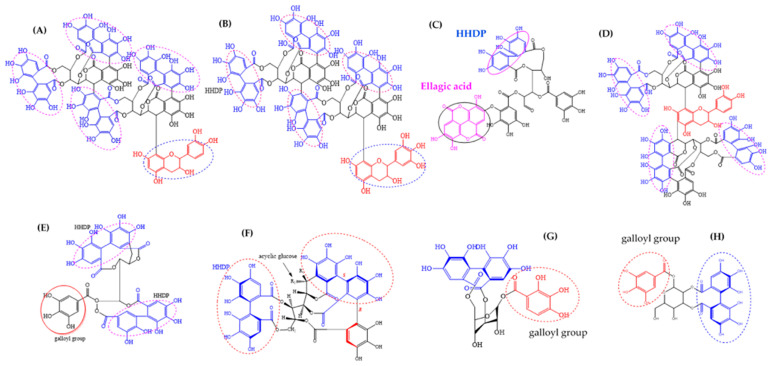
Chemical structure of ellagitannins identified from *Anogeissus* spp. and *A. leiocarpa*: (**A**) flavano-ellagitannin anogeissinin A, (**B**) anogeissinin B, (**C**) cornusiin B, (**D**) anogeissinin, (**E**) casuarinin, (**F**) castalagin (R1 = H; R2 = OH) and vescalagin (R1 = OH; R2 = H), (**G**) corilagin, and (**H**) isomer of corilagin sanguiin H4. DHHDP: dehydro-hexahydroxydiphenoyl; HHDP: hexahydroxydiphenoyl. Source for molecular structures: CAS, 2020.

**Table 1 antibiotics-09-00364-t001:** In vitro antimycobacterial activity against *Mycobacterium smegmatis* ATCC 14468 of extracts and fractions of various polarities from *Anogeissus leiocarpa* roots, stem bark, and leaves. Results obtained with an agar diffusion method. Activity index (AI) was calculated in relation to rifampicin.

Extracts of *Anogeissus leiocarpa*	IZD	SEM	AI
**R. MeSox**	**21.33**	0.33	**0.45**
R. acetone	15.45	0.33	0.32
R. hex	NA		
**R. EtOAc**	**28.50**	0.29	**0.60**
R. aqu	18.67	0.17	0.39
R. Dic	16.00	0.00	0.34
R. H_2_O*	18.67	0.17	0.39
**R. Me***	**21.33**	0.17	**0.45**
R. HH_2_O	13.30	0.17	0.28
Rb. acetone	18.33	0.17	0.39
Rb. Me*	17.67	0.17	0.37
Rb. Dic	16.33	0.17	0.34
Rb. H_2_O*	18.67	0.17	0.39
Rb. hex	NA		
**W. MeSox**	**20.17**	0.33	0.42
W. hex	NA		
W. H_2_O*	NA		
W. CHCl_3_	13.67	0.17	0.29
W. HH_2_O	15.00	0.00	0.32
W. aqu	14.00	0.00	0.29
**W. EtOAc**	**25.17**	0.17	**0.53**
**W. Me***	**20.33**	0.33	**0.40**
**B. EtOAc**	**28.67**	0.17	**0.60**
**B. aqu**	**25.00**	0.00	**0.53**
B. MeSox	14.83	0.17	0.31
B. hex	NA		
B. CHCl_3_	NA		
B. H_2_O*	17.67	0.17	0.37
B. Me*	18.67	0.17	0.39
**B. HH_2_O**	**20.67**	0.17	**0.44**
**L. EtOAc**	**30.50**	0.29	**0.64**
**L. aqu**	**24.67**	0.17	**0.52**
**L. H_2_O***	**26.67**	0.17	**0.56**
L. hex	NA		
**L. Me***	**20.00**	0.29	**0.42**
**L. acetone**	**19.94**	0.29	**0.42**
L. Dic	NT		
L. MeSox	**24.17**	0.17	**0.51**
L. HH_2_O	**20.33**	0.33	**0.43**
**Rifampicin**	**47.5**	**0.29**	**1.00**
**Methanol**	**NA**		
**Hexane**	**NA**		

W, stem wood; B, stem bark; R, root; L, leaves; Rb, root bark; Me*, cold methanol extracts; EtOAc, ethyl acetate extracts; hex, hexane extracts; Dic, dichloromethane; CHCl_3_, chloroform extracts; aqu, aqueous extracts; HH_2_O), hot water decoctions; MeSox, methanolic Soxhlet extracts; H_2_O^*^, cold water extracts; NA, not active; and NT, not tested; AI, activity index, which was calculated in relation to the antibiotic rifampicin; IZD, diameter of inhibition zones recorded in mm as the mean of triplicates (*n* = 3) ± SEM of three experiments; methanol, and hexane were used as solvent control; most promising results (IZD ≥ 20 mm) are indicated by bold text. Filter paper disks (⌀ 12.7 mm) were saturated with 200 μL extracts/fractions (50 mg/mL) and rifampicin (10 mg/mL).

**Table 2 antibiotics-09-00364-t002:** Minimum inhibitory concentration of the extracts and fractions of *A. leiocarpa* against *Mycobacterium smegmatis* ATCC 14468.

*A. leiocarpa* Crude Extracts and Fractions	Total Activity (in mL/g)	MIC (in µg/mL)
W. MeSox	40.2	5000 **
W. Me*	4.2	5000
**B. EtOAc**	21.6	**2500** **
B. aqu	16	5000
B. HH2O	13.6	5000
**R. MeSox and its preparative** **reversed phase 18** **(RP_18_)-** **thin layer chromatography** **(TLC) fractions (Fr3-Fr8)**	43.2	**2500**
**Fr 3 (Rf 0.095)**		**1500 (IC 89) ****
**Fr 4 (Rf 0.159)**		**1000 (IC 89) ****
**Fr 6 (Rf 0.276)**		**2000 (IC 91) ****
**Fr 8 (Rf 0.457)**		**3000 (IC 90) ****
R. Me*	5.6	5000
**R. EtOAc**	7.04	**625** **
L. MeSox	101.6	5000
L. HH2O	14.8	5000
**L. H2O***	28.4	**2500**
**L. EtOAc**	38	**2500**
**Pure compounds known to be present in *A. leiocarpa***		
**Gallic acid**		**500 (IC 98) ****
**Quercetin**		**250 (IC 94) ****
**Apigenin**		**250 (IC 97) ****
**Corilagin**		**1000 (IC 94) ****
**Ellagic acid**		**500 (IC 98) ****
**Rifampicin**		39.06 µg/mL (3.90 µg/mL, IC 98) **

** Results obtained with a microplate method. Other results not marked with ** were obtained with an agar diffusion method. W, stem wood; B, stem bark; R, root; L, leaves; Me*, cold methanol extracts; EtOAc, ethyl acetate extracts; HH_2_O, hot water extracts or decoctions; MeSox, methanolic Soxhlet extracts; H_2_O*, cold water extracts, macerations; total activity in mL/g; IC, inhibitory concentration that indicates the % of growth inhibition compared to the growth control at that concentration; **Fr_3_**, **Fr_4_**, **Fr_6_**, and **Fr_8_** were obtained from R. MeSox using RP_18_-TLC.

**Table 3 antibiotics-09-00364-t003:** HPLC-DAD, UHPLC/QTOF-MS and RP_18_ TLC data of fractions obtained from a methanol Soxhlet root extract of *A. leiocarpa*. Some compounds numbered according to the HPLC-DAD chromatograms in Figure 5.

RP_18-_Thin Layer Chromatography Fractions and Their Compounds	Molecular Formula	Rt HPLC-DAD	Rt LC-MS	[M-H]^−^	Exact Calculated MW	UV_λ_ max (HPLC-DAD)	Peak Area %	Distance Moved in cm on TLC Plate	RP_18_ TLC R_f_ Value	DPPH Reactive
**Fr_3_ MeSox**								1.2	0.0945	Yes
Ellagic acid derivative		20.04				254, 366	2.71			
Di-methyl ellagic acid xyloside (**18**)	C_21_H_18_O_12_	23.47	9.65	461.0739	462.0739	246, 376	**4.93**			
Di-methyl-ellagic acid (**21**)	C_16_H_10_O_8_	28.09	11.25	329.0318	330.0318	246, 380	**38.43**			
Ellagic acid derivative		29.94				248, 374	**8.05**			
Unknown		33.06				216, 248, 374	**4.31**			
Acetylated ellagic acid derivative (**24**)		35.08	12.80	343.0477	344.0477	222, 246, 370	**5.99**			
**Fr_4_ MeSox**								1.9	0.1496	Yes
Protocatechuic acid (**2**)	C_7_H_6_O_4_	3.69	1.34	153.0196	154.0196	218, 220, 260, 294	1.5			
Ellagic acid derivative		18.12				246, 370	1.63	
Ellagic acid derivative		18.90				210, 254, 362	1.7			
Ellagic acid derivative		19.98				210, 254, 368	**19.12**			
Di-methyl ellagic acid xyloside (**18**)	C_21_H_18_O_12_	23.52				246, 376	**9.09**			
Di-methyl ellagic acid (**21**)	C_16_H_10_O_8_	28.07	11.25	329.0318	330.0318	246, 380	**21.92**			
Ellagic acid derivative		35.08				246, 382	3.18			
**Fr_6_ MeSox**								3.5	0.2756	Yes
Unknown flavonoid (**9a**)		14.66				232, 276, 350	2.31			
Ellagic acid derivative		17.54				254, 378	**9.84**			
Ellagic acid derivative		18.12				246, 370	**13.41**			
Unknown		18.93				254, 360	**7.77**			
Ellagic acid derivatives		19.99				254, 362	**22.09**			
ellagitannin (unknown)		23.64				220, 248, 378	2.92			
Di-methyl ellagic acid (**21**)	C_16_H_10_O_8_	29.01	11.25	329.0318	330.0318	248, 376	3.99			
**Fr_8_ MeSox**								5.8	0.4567	Yes
Unknown		5.057				242	3.65			
Unknown		13.07				210, 230, 282	3.99			
Ellagic acid derivative		17.52				254, 384	3.96			

(Fr), fraction; (DMF), distance moved by fraction on the RP_18_-TLC plate; DPPH, 2,2-diphenoyl-1picrylhydrazyl reagent measuring antioxidant activity; and MW, exact calculated mass that was calculated from the molecular formula.

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
