# Peer review of "Potential Anti-Tuberculosis Activity of the Extracts and Their Active Components of Anogeissus leiocarpa (DC.) Guill. and Perr. with Special Emphasis on Polyphenols"

_antibiotics, 2020, doi:10.3390/antibiotics9070364_

Round 1

Reviewer 1 Report

Journal: Antibiotics

Manuscript ID: antibiotics-831580
Type of manuscript: Article
Title: Antimycobacterial activity of polyphenol-rich extracts of the African medicinal plant Anogeissus leiocarpus (DC.) Guill. & Perr.

Authors: Enass Y. A. Salih, Riitta Julkunen-Tiitto, Olavi Luukkanen, Marketta Sipi, Mustafa K. M. Fahmi, Pia Johanna Fyhrquist

In the present investigation, the extracts of various parts of Anogeissus leiocarpus (DC.) Guill. & Perr (Combretaceae) were tested for their growth inhibitory effects on Mycobacterium smegmatis ATCC 14468, which showed that several of these extracts were active, and the ethyl acetate extract of the roots of A. leiocarpus exhibited the most potent activity. To identify the active components, a preparative RP-18 thin-layer fractionation was developed and performed, which showed that ellagic acid derivatives could be the major active components, including di-methyl ellagic acid and its xyloside, and thus the roots, stem bark, and leaves of A. leiocarpus could be used in Sudanese folk medicine as a maceration against cough related to TB. This manuscript indicates that ellagic acid derivatives could be regarded as promising lead compounds for future discovery of new anti-tuberculosis drugs. However, the authors seem not to test any of these compounds in their investigation, and thus this manuscript is recommended to be re-considered as a potential Article published in Antibiotics. The title could be revised as “Potential anti-tuberculosis activity of the extracts and their active components of Anogeissus leiocarpus”, and the abstract should be shorten greatly, while the anti-tuberculosis activity should be tested for both di-methyl ellagic acid and di-methyl ellagic acid xyloside, with the results of testing these compounds and the positive and negative controls included in the present manuscript.

Author Response

General revision of the manuscript:

  1. Requested changes in the English language were made and marked with the red colour in the manuscript.
  2. The whole manuscript has been shortened and improved. The discussion part is now more focused on discussing results obtained in this paper in relation to results reported in the literature. The introduction part and abstract part as well as the results part were shortened and written in a more focused way.
  3. The research design was improved and marked with the red color in the manuscript. )
  4. Some formulas and explanations for calculations have been added to the method part, such as the calculation of the total antimicrobial activity. Unnecessary text was removed from the method part.

Response to Reviewer 1:

Comment 1: The authors seem not tested di-methyl ellagic acid and its xyloside or any of the investigated compounds in this work, and thus this manuscript is recommended to be re-considered as a potential Article published in Antibiotics.

Author`s response: We agree with the reviewer that di-methyl ellagic acid and its xyloside could have been tested for their MIC values in our antimycobacterial test system. However, since there already exists information on the antimycobacterial effects of ellagic acid xyloside in the literature, and it has been found to inhibit both Mycobacterium smegmatis and M. intracellulare with a MIC of 4.88 µg/ml, we found that it was unnecessary to re-test this compound. From this cited research (Kuete et al., 2010, reference no. 39 in our paper) and a study on the docking capacity of pteleoellagic acid to an important enzyme critical to the synthesis of mycobacterial cell wall lipid (Shilpi et al., 2015 reference no. 44), we could draw the conclusion that ellagic acid based compounds, and especially their glycosides could be prospective new antimycobacterial drug lead compounds. We indeed think that it could be interesting to isolate various ellagic acid derivatives from Anogeissus leiocarpus and to test them and their combinations on mycobacterial strains. However, for this isolation purpose a very large amount of starting material (in kilos), will be needed to get sufficient material for antimycobacterial testing. Thus, this was not an option for us at this stage. At the moment, due to the covid-19 situation, it is not possible for us to order the suggested ellagic acid derivatives and nor to perform extra testing in the lab. However, we have now, in the improved manuscript included ellagic acid in order to have one ellagic acid reference. Please see the antimycobacterial result for this compound both in the text in lines 134-139 and in Table 2 as well as in the discussion part concerning the potential of ellagic acid derivatives as antimycobacterial compounds (Lines 342-379). We apologize for this inconvenience that our study was lacking this important ellagic acid reference in the previous version of our manuscript- we simply forgot to add it in our paper. It has been suggested that the xylose part in ellagic acid derivatives play an important role in improving their activity against Mycobacterium spp. Our result on the activity of ellagic acid showing a mild growth inhibitory effect against M. smegmatis (MIC 500 µg/ml) compared to the reported MIC in literature of dimethyl ellagic acid xyloside against M. smegmatis (4.88 ug/ml) (Kuete et al., 2010; reference no. 39), supports this theory, that a conjugated sugar molecule would be needed to increase the activity. In addition to ellagic acid, we have also included corilagin as a reference in Table 2 in our revised manuscript.

Comment 2: The title could be revised as “Potential anti-tuberculosis activity of the extracts and their active components of Anogeissus leiocarpus”, and the abstract should be shorten greatly, while the anti-tuberculosis activity should be tested for both di-methyl ellagic acid and di-methyl ellagic acid xyloside, with the results of testing these compounds and the positive and negative controls included in the present manuscript.

Author`s response: The suggested title by reviewer 2 “Potential anti-tuberculosis activity of the extracts and their active components of Anogeissus leiocarpa” is good and we have now changed the title accordingly.

The abstract length has been adjusted to 265 words.

Since di-methyl ellagic acid and its xyloside have been tested for their growth inhibitory effects against M. smegmatis and M. intracellulare (Kuete et al., 2010, reference no. 39 and Kaneko & Kaneko, 2013, reference no. 41) we felt that we do not need to repeat this.

The solvent controls, methanol and hexane, have been added to table 1 and highlighted with the red color.

Reviewer 2 Report

First, I would like to ask you to clarify a point - there are two taxonomic names in scientific literature -Anogeissus leiocarpus or leiocarpa- do they refer to the same organism / plant?  If so, please explain why you use A.leiocarpus.

The Introduction is well written. But,I think that figure 2 is completely unnecessary - the information, also mentioned in the text (avoid unnecessary repetitions), is unnecessary for the perception of the content of the article. The mention in the text is enough.

Results section The results description contains a lot of discussion elements - please move them to the Discussion section. In general, this part is written in a very detailed and haotic way - too much (often irrelevant) information - so you get the impression that the authors had a problem with the selection of the obtained experimental results and therefore showed everything - how it goes. This attitude is a sign of scientific immaturity. Shortening the description of the results is absolutely necessary to improve the image of the publication and enable it to be published.

I suggest moving some of the less important information to the supplement. which will allow a more discerning reader to become acquainted with them and not introduce the impression of haos.

The description of the methods used is also very detailed - I suggest simplifying it by removing less relevant information.

I think that after introducing these amendments the publication will gain in value.

Author Response

Comment 1: Does Anogeissus leiocarpus and leiocarpa refer to the same plant and explain why the authors use A. leiocarpus?

Author`s response: We have investigated this and found that reviewer 2 is right. The latest accepted scientific name according to The Plant List database in 2012 (Royal Botanic garden, Kew and Missouri botanical garden), is Anogeissus leiocarpa (DC.) Guill. & Perr and A. leiocarpus is a synonym. Therefore, we have changed A. leiocarpus to A. leiocarpa in the manuscript.

Comment 2: The Introduction is well written. But, figure 2 is completely unnecessary. The information, also mentioned in the text (avoid unnecessary repetitions), is unnecessary for the perception of the content of the article.

Author`s response: We thank you for praising our written introduction part. However, we would like to justify the importance of figure 2 as this figure gives a graphical clarification of the number of completed research papers in A. leiocarpa since 1998. Moreover, the figure 2 quickly illustrates for the reader that the number of papers regarding the phytochemical composition of A. leiocarpa are very few. Thus we would like to keep figure 2.

Comment 3: The results description contains a lot of discussion elements, please move them to the Discussion section. In general, this part is written in a very detailed and chaotic way - too much (often irrelevant) information, so you get the impression that the authors had a problem with the selection of the obtained experimental results and therefore showed everything, how it goes. This attitude is a sign of scientific immaturity. Shortening the description of the results is absolutely necessary to improve the image of the publication and enable it to be published.

I suggest moving some of the less important information to the supplement. Which will allow a more discerning reader to become acquainted with them and not introduce the impression of chaos.

The description of the methods used is also very detailed. I suggest simplifying it by removing less relevant information.

Author`s response: Thank you for your suggestion to shorten the results description and to move discussion elements to discussion part. We have tried to improve and to shorten both the results and discussion parts. The results part has been made denser. In the discussion part, we have now discussed the main findings in our research in light of the research of other authors. Our results on the total activity are discussed and in the discussion part, we focus mainly on ellagic acid derivatives, ellagitannins and flavonoids that others and we have found in A. leiocarpa and their role as possible antimycobacterial compounds in our extracts that showed good antimycobacterial effects in this investigation. We did an immense effort on this manuscript and the information and results should provide a key for future researchers to benefit from these kind of results and information. In the light of the fact that this is the first investigation on the antimycobacterial effects of extracts, fractions and some compounds of A. leiocarpa, the results are interesting and definitely provide an important contribution for this research field.

Reviewer 3 Report

The authors report the results on the antimycobacterial activity of extracts of Anogeissus leiocarpus against Mycobacterium smegmatis. The results are very interesting and could provide an important contribution on this research field. However, the presentation of the manuscript is very low.

General comments

  • Manuscript presentation should be revised thoroughly for text style, section/paragraph organization, spacing and tables presentation.
  • L42, 207, 255, 258, 259, 261, 262, 326, 337, 458, 636, 638, 639, 670 – “antioxidant” instead of “antioxidative”.
  • Figures 5 and 6 could also be presented in Supplementary material.

Specific comments

Abstract

The abstract should be shortened (200 words maximum). Avoid the use of abbreviations.

Introduction

The introduction should be shortened in lines 88-135.

Results

  • Manuscript should contain:
  • HPLC quantitative analysis of investigated compounds.
  • Statistical analysis.

Providing these results will add scientific soundness and confirm some general statements provided (e.g. L41, L44,).

  • L191-193 authors should present the results of total activity in Table 2 (Also, authors should present calculations in material and methods section).
  • Result section contains some discussion sentences that should be moved in discussion section (g. L185-205; 229-263). In alternative authors could merge these sections.

Discussion

Authors correctly discussed the results in perspective of previous studies and of the working hypotheses. Limitations in the use of natural extracts should be included (e.g. secondary metabolites concentration variability in plants and extracts). This section could be merged with results.

Material and Methods

Provide extraction yields calculations.

Conclusion

Remove bullet points, organizing section in one single paragraph.

References

Citation in the text should be revised, remove years and “a” and “b” from references with same author and year. Reference list should be revised in both style and content. Some examples:

  • L969 (Journal name in italics)
  • L974 (remove second “14(1), 400”)
  • Remove “pp” before pages ranges of journal articles.

Author Response

Comment 1: The results are very interesting and could provide an important contribution on this research field. However, the presentation of the manuscript is very low.

Author`s response: According to your wishes, we have tried to improve our manuscript now, so that the results stand out more clearly.

Comment 2: Manuscript presentation should be revised thoroughly for text style, section/paragraph organization, spacing and tables presentation.

L42, 207, 255, 258, 259, 261, 262, 326, 337, 458, 636, 638, 639, 670 – “antioxidant” instead of “antioxidative”.

Author`s response: We have adjusted the manuscript according to the reviewer wishes. The word antioxidative is replaced by antioxidant and highlighted with the red color now in lines 32, 151, 179, 182, 231, 246, 338, 341, 456, 526, 528, 529, and 557.

Comment 3: Figures 5 and 6 could also be presented in Supplementary material.

Author`s response: We think that Figures 5 and 6 should appear close to the text where they are discussed, to facilitate the understanding of the text. In addition, figure 5 explained important chromatograms of our tested TLC fractions in relation to their mother crude extract. Also, as we response to reviewer no. 4 in his Comment 4, that figure 6, illustrates a summary of the structures of some of the chemical compounds that we or other researchers detected in the genus Anogeissus and/or in A. leiocarpa. This figure connects very well to our introduction, result and discussion parts. Therefore, we think that it is very important to keep both figures to facilite the concept of the article.

Comment 4: The abstract should be shortened (200 words maximum). Avoid the use of abbreviations.

Author`s response: The abstract was shortened according to the guidelines of manuscript preparation for Antibiotics and now contains 265 words, describing briefly the main methods, summarizing the article's main findings and outlining the main conclusions.

We think that the abbreviations such as MIC and IZD are important in the abstract. However, for IZD, inhibition zone diameter, which is less known, we have also written out the whole meaning before the abbreviation.

Comment 5: The introduction should be shortened in lines 88-135.

Author`s response: We would like to keep this very important information in the introduction part, such as the ethnopharmacological uses and geographical occurrence as well as brief botanical description of the studied species, Anogeissus leiocarpa. Also the explanation on the previous research done on Anogeissus leocarpa as a medicinal plant against infectious diseases, as well as the earlier identified chemical constituents in this species is very essential information for the readers of our paper, in order to avoid research repetition. Therefore, we do not agree with the reviewer 3 that the introduction text of these specific parts should be shortened.

Comment 6: Manuscript should contain: HPLC quantitative analysis of investigated compounds and statistical analysis.

Providing these results will add scientific soundness and confirm some general statements provided (e.g. L41, L44,).

Author`s response: We agree with the reviewer. Our manuscript contained HPLC-DAD (peak area %) and mass-spectrometric quantitative analysis UHPLC/QTOF-MS. We have previously analyzed the crude extracts (figure 5 A) using HPLC-DAD and UHPLC/QTOF-MS (Salih et al 2017a). In this manuscript, we have separated fractions (figure 5 B) from the crude extracts (figure 5 A) using thin layer chromatography (RP-18 TLC). These TLC fractions have been analyzed in HPLC-DAD and compared to the corresponding chemical constituents in the crude methanol extract (in figure 5 A) using UHPLC/QTOF-MS. The quantitative composition of ellagic acid derivatives and other compounds in these TLC-fractions is shown in Table 3 as peak area %.

We have used basic statistical analysis in this manuscript: The antimycobacterial assays were done in triplicate and the standard error of means (SEM) were calculated. Besides, the % bacterial growth and % of growth inhibition ±SEM were carefully calculated.

Comment 7: L191-193 authors should present the results of total activity in Table 2 (Also, authors should present calculations in material and methods section).

Author`s response: We have added the total activity of A. leiocarpa extracts in table 2 (highlighted in red color). Moreover, the formula for the calculation of the total activity has been added to the material and method part on lines 671-672.

Comment 8: Result section contains some discussion sentences that should be moved in discussion section (g. L185-205; 229-263). In alternative authors could merge these sections.

Author`s response: We have moved sentences from the results part that would fit better in the discussion part to the discussion part.

Since the article was originally written with a separate results and discussion part, it would be difficult to merge these parts. Thus, we will keep these parts separate.

Comment 9: Authors correctly discussed the results in perspective of previous studies and of the working hypotheses. Limitations in the use of natural extracts should be included (e.g. secondary metabolites concentration variability in plants and extracts). This section could be merged with results.

Author`s response: We agree with reviewer, there are limitations for using natural extracts as sources of biologically active compounds due to concentration variability between different source plant individuals (and different extracts). This has now been discussed with a few lines (lines 332-335) in the discussion part.

Comment 10: Provide extraction yields calculations.

Author`s response: Extraction yields formula was added to material and method part and highlighted with the red color.

Percentage of Extraction Yield (%) = weight of the dry extract / weight of dry plant material before extraction x 100.

We have also placed the percentage yield in Y axis in Fig 3 according to the wish of reviewer no. 4.

Comment 11: Remove bullet points, organizing conclusions section in one single paragraph.

Author`s response: According to Antibiotic guidelines, conclusions part is optional and not mandatory “conclusion briefly outlining the take-home message and the lessons learned”. Therefore, we thought that it is better to keep the bullet points since it makes the conclusion part more understandable. We have therefore chosen to keep the bullets.

Comment 12: Citation in the text should be revised, remove years and “a” and “b” from references with same author and year. Reference list should be revised in both style and content. Some examples:

  • L969 (Journal name in italics)
  • L974 (remove second “14(1), 400”)
  • Remove “pp” before pages ranges of journal articles.

Author`s response: Thank you for these observations. The mentioned citations now have been revised in the text. However, to differentiate between various articles written by the same author in the same year, we prefer to use the symbols a and b or any another symbol. Reference journal name in italics have been fixed in the manuscript. Also we removed second “14(1), 400”.

We agree with the reviewer that the “pp” before pages ranges of journal articles have to be removed and now it has been removed from all journal article references. The new references were highlighted in red color in the reference part.

Reviewer 4 Report

This is an interesting work, especially due to the fact that it presents a possible  use of medicinal plant in treatment of TB in the difficult area of ​​South Sudan, with limited access to treatment and with high morbidity. However, this work has a number of weaknesses. It is chaotic and lengthy. Especially, Introduction contains a lot of statistical and botanical information that is not useful to the reader. In my opinion, authors should focused on the presentation of results obtained from experiments using i.e. TLC, HPLC-UV/DAD, UHPLC/Q, microbiological analysis, and show these results in relation to potential antimycobacterial use.

The work should be rewritten and re-submitted.

Additionally:

Abstract: The abstract should be a total of about 200 words maximum and should be restricted to  the purpose of the study; main methods,  main findings, and conclusion  or interpretations.

Introduction: is too long with many not necessary information, figure 2 is redundant.

Results: Fig 3 A, B, C lack description of Y axis. Are figures 6 really necessary?

Materials and Methods: line 587 please specify what plant

Line 601 a mistake: “Germany”

Conclusions: line 787; “our results justify” on what basis,  were the bioavailability tests of the obtained preparations carried out ?

Author Response

Comment 1: This is an interesting work, especially due to the fact that it presents a possible use of medicinal plant in treatment of TB in the difficult area of ​​South Sudan, with limited access to treatment and with high morbidity. However, this work has a number of weaknesses. It is chaotic and lengthy. Especially, Introduction contains a lot of statistical and botanical information that is not useful to the reader. In my opinion, authors should focused on the presentation of results obtained from experiments using i.e. TLC, HPLC-UV/DAD, UHPLC/Q, microbiological analysis, and show these results in relation to potential antimycobacterial use.

Author`s response: First, thank you to the fourth reviewer. We have now tried to reduce the length of the manuscript. The introduction part has been shortened. More focus has now been put on presenting TLC, HPLC-UV/DAD and UHPLC/Q-TOF results, both in the results part and in relation to other author’s work in the discussion part.

Moreover, some specific details, for example those dealing with the supposed antibacterial mechanisms of action of stilbenes, are removed from the manuscript.

We would like to justify the botanical information of Anogeissus leiocarpa in our introduction part. Correct botanical information is extremely important in medicinal plant research. Thus, we have chosen not to remove the part describing botanical details on A. leiocarpa.

In general, we observed that, in this kind of scientific research, there are weaknesses, such as omitting the botanical description of the medicinal plant species studied. Correct botanical information works as a base for other researchers to add their studies on that/those medicinal plant(-s).

Comment 2: The abstract should be a total of about 200 words maximum and should be restricted to the purpose of the study; main methods, main findings, and conclusion  or interpretations.

Author`s response: We have now shortened our abstract to 265 words without deleting important information.

Comment 3: Introduction: is too long with many not necessary information, figure 2 is redundant.

Author`s response: The introduction is long since our article deals and presents both the antimycobacterial activities of various extracts and fractions of A. leiocarpa as well as identification of secondary compounds of those extracts. As we responded to reviewer no. 3, in the introduction part we have presented the previous research done in Anogeissus leiocarpa as a medicinal plant against infectious disease (in vitro antimicrobial testing). In addition, in the introduction part we present earlier identified chemical constituents in this species. This is very essential to avoid research repetition in order to be able to go on with novel research work. Therefore, we have tried to densify the introduction part without leaving away those important aspects discussed above.

We would like to keep figure 2, as it illustrates nicely the research that has been made on A. leiocarpus from the 1990s onward. The figure nicely demonstrates the need for more research on this interesting medicinal plant; both when it comes to its phytochemistry (almost no alkaloids have been detected from this so far with the exception of kynurenic acid) and its potential as a source of antimicrobial lead compounds and antibiotic adjuvants.

Comment 4: Fig 3 A, B, C lack description of Y axis. Are figures 6 really necessary?

Author`s response: We have added the percentage of extraction yield to the Y axis in Fig 3 A, B, C. However, we would like to justify the reason why Figure 6 is necessary to the manuscript: this figure illustrates a summary of the structures of some of the chemical compounds that we or other researchers detected in the genus Anogeissus and/or in A. leiocarpa. This figure connects very well to our introduction, result and discussion parts. In addition, the chemical structures of compound classes found in A. leiocarpa are rarely presented in one paper like this. We think that it is very important that these molecular structures are presented here to be available for all researchers around the globe in this open access journal.

Comment 5: line 472 please specify what plant? Line 485 a mistake: “Germany”

Author`s response: We have specified the name of the plant of Anogeissus leiocarpa in line 472 now, and highlighted with red color. The word “Germany” is corrected and highlighted with the red color on line 485 now.

Comment 6: Conclusions: line 677; “our results justify” on what basis, were the bioavailability tests of the obtained preparations carried out?

Author`s response: We did not test the bioavailability. This should be done in the future, but facilities for this are not available in our laboratory at the moment. Our in vitro results could however give some information for the validation of the traditional oral application of utilizing water extracts and decoctions of A. leiocarpus roots, leaves and stem bark to treat cough related to TB. Thus, we have modified the sentence in our conclusions part, line 677 to: “Our results could justify the traditional oral application of utilizing water extracts and decoctions of A. leiocarpa roots, leaves and stem bark to treat cough related to TB”, with the underlined word “could” inserted in this sentence.

When it comes to the toxicity of water extracts of A. leiocarpa, it was found that longtime use of this species against TB in Sudanese traditional medicine, did not result in negative symptoms or side effects. This was observed both during and after the medication of the TB patients. However, as the reviewer implies, more research is needed.

Round 2

Reviewer 1 Report

Journal: Antibiotics

Manuscript ID: antibiotics-831580
Type of manuscript: Article
Title: Potential anti-tuberculosis activity of the extracts and their active components of Anogeissus leiocarpa (DC.) Guill. & Perr. with special emphasis on polyphenols

Authors: Enass Y. A. Salih, Riitta Julkunen-Tiitto, Olavi Luukkanen, Marketta Sipi, Mustafa K. M. Fahmi, Pia Johanna Fyhrquist

In the present investigation, the extracts of various parts of Anogeissus leiocarpus (DC.) Guill. & Perr (Combretaceae) were tested for their growth inhibitory effects on Mycobacterium smegmatis ATCC 14468, which showed that several of these extracts were active, and the ethyl acetate extract of the roots of A. leiocarpus exhibited the most potent activity. To identify the active components, a preparative RP-18 thin-layer fractionation was developed and performed, which showed that ellagic acid derivatives could be the major active components, including di-methyl ellagic acid and its xyloside, and thus the roots, stem bark, and leaves of A. leiocarpus could be used in Sudanese folk medicine as a maceration against cough related to TB. This manuscript indicates that ellagic acid derivatives could be regarded as the major active components of A. leiocarpus as promising lead compounds for future discovery of new anti-tuberculosis drugs. However, the authors seem not to identify any of these compounds in their investigation, and the compounds presented in Figure 6 seem a mini-review, with no sufficient data to support their structural elucidation. Therefore, this manuscript could be rejected for a potential Article published in Antibiotics.

Reviewer 2 Report

Now is aceptable.

Reviewer 3 Report

The authors addressed and responded correctly to my previous comments and suggestions. I have no further suggestions.

Reviewer 4 Report

The manuscript may be published in present form. The authors' responses to review are satisfying for me.